# CLIP FACIAL EXPRESSION RECOGNITION: BALANCING PRECISION AND GENERALIZATION

## ABSTRACT

Current facial expression recognition (FER) methods excel in achieving high classification accuracy but often struggle to generalize effectively across various unseen test sets. On the other hand, CLIP (Radford et al., 2021) demonstrates impressive generalization ability, albeit at the cost of lower classification accuracy compared to SOTA FER methods. In this paper, we propose a novel approach to adapt CLIP for FER, striking a balance between precision and generalization. Our motivation is rooted in the potential of large pre-trained models like CLIP to extract generalizable face features across diverse FER domains, showcasing high generalization ability. However, these extracted face features, which include extra information like age and gender, are not directly suitable for FER tasks, resulting in lower classification accuracy. To solve this problem, we train a traditional FER model to learn sigmoid masks to only select expression-related features from the fixed CLIP face features. The selected features are utilized for classification. To improve the generalization ability of the learned masks, we propose a channel-separation module to map the channels of the masked features directly to logits and avoid using the FC layer to reduce overfitting. We also introduce a channel-diverse loss to make the learned masks as diverse as possible. Extensive experiments on numerous FER datasets verify that our method outperforms SOTA FER methods by large margins. *Based on both the high classification accuracy and generalization ability, our proposed method has the potential to become a new paradigm in the FER field.* The code will be available.

## 1 INTRODUCTION

Facial expression recognition (FER) aims to understand human feelings and is vital to human-computer interaction. With the development of deep learning, FER methods achieve high classification accuracy on both laboratory-collected and in-the-wild FER datasets. However, we find that things are different when it comes to the generalization ability of these SOTA FER methods. For example, if we train the FER model on a given FER dataset like RAF-DB (Li et al., 2017), then it can only achieve high classification accuracy on the test set of RAF-DB, while very low classification accuracy on other different FER test sets like AffectNet (Mollahosseini et al., 2017). Though some works try to deal with the domain generalization problem in FER, they assume accessing labeled or unlabeled target domain samples. However, in real-world FER, as we do not previously know the distribution of the target domain, we might not even have access to unlabeled target samples. In such cases, the domain adaptation FER methods cannot work. In this paper, we aim to study the balance of classification accuracy and generalization ability of SOTA FER methods. It is well known that high classification accuracy does not always mean high generalization ability. Through experiments, we find the opposite is more likely to be true. The baseline method could outperform SOTA FER methods on test sets with domain gaps of the train set.

Existing FER methods fail to directly generalize to other test sets, we speculate that these methods might fit the given train set too well to predict only based on expression features on unseen test sets. Inspired by the strong zero-shot knowledge transfer of CLIP (Radford et al., 2021) across different downstream tasks, we try to adapt it for FER methods to improve their generalization ability. Our assumption is that CLIP is trained using large-scale image-text pairs, thus, it generalizes well across different FER domains. However, the classification accuracy of CLIP-based models is low in the FER task as the features extracted by CLIP contain much more information that is

Figure 1: To evaluate the generalization ability of FER methods, we train FER models on one train set and test the trained models on other unseen test sets. SOTA FER methods do not work well on unseen test sets, which are unreliable in real-world deployment. Through the learned sigmoid masks, channel-separation module, and channel-diverse loss, we adapt CLIP to FER and outperform existing SOTA FER methods like EAC, by large margins on different FER test sets.

not related to expression such as age, gender, race, etc. To solve this problem, we design a novel method to guide the FER model to learn representations that only relate to facial expressions across faces from different domains. Specifically, we first extract generalizable face features from CLIP and fix them during the training to maintain the generalization ability. We train another FER model to learn masks based on the fixed face features from CLIP. After combining the learned masks with the fixed face features, the expression features are extracted. In order to make the learned masks generalizable instead of overfitting the given images, we cleverly utilize a sigmoid function to regularize the learned masks. Furthermore, we propose a channel-separation module to separate the learned masks into pieces according to expression classes. This operation avoids using the FC layer and directly maps different channels to logits for different categories, which further reduces the overfitting probability of the FER model. Finally, we introduce a channel-diverse loss to learn the masks corresponding to different expressions as diverse as possible.

Extensive experiments on five different FER datasets validate the effectiveness of our proposed method. To the best of our knowledge, our work is the first to adapt CLIP to improve the generalization ability of FER methods. We summarize our main contributions as follows.

- We investigate the trade-off between classification accuracy and generalization ability in the facial expression recognition field, which is relatively less touched as SOTA FER methods mainly focus on improving classification accuracy.
- We design a novel method to adapt CLIP for facial expression recognition. We propose to learn sigmoid masks on fixed face features extracted by CLIP to detect expressions. We further propose a channel-separation module and a channel-diverse loss to increase the generalization ability of the learned masks.
- Extensive experiments on five different FER methods illustrate that our method outperforms SOTA FER methods by remarkable margins. Our method achieves both high classification accuracy and high generalization ability.

## 2 RELATED WORK

### 2.1 FACIAL EXPRESSION RECOGNITION

Facial Expression Recognition (FER) plays a vital role in human-computer interaction, and extensive research has been conducted to enhance the precision of FER (Shan et al., 2009; Zhi et al., 2010; Zhong et al., 2012; Bargal et al., 2016; Kahou et al., 2013; Farzaneh & Qi, 2021; Ruan et al., 2021; Li et al., 2017). For instance, Li *et al.* (Li et al., 2017) use crowd-sourcing to simulate human expression recognition, while (Bargal et al., 2016; Kahou et al., 2013) employ model ensembling to leverage more information. Farzaneh *et al.* (Farzaneh & Qi, 2021) propose a center loss variant to maximize intra-class similarity and inter-class separation for FER, and Ruan *et al.* (Ruan et al., 2021) acquire expression-relevant information during the decomposition of an expression feature. However, we find that these FER methods are effective when the test set has no domain gap with the train set, while their performance drops drastically when facing domain-different test sets. In

this paper, we aim to improve the generalization ability of FER methods and maintain their high classification accuracy to make them suitable for real-world deployment.

## 2.2 DOMAIN GENERALIZATION

Domain Generalization (DG) aims to help models trained on a set of source domains generalize better on unseen target domains. A common practice is to reduce the feature discrepancy among multiple source domains. (Tzeng et al., 2014; Long et al., 2015; 2017) all adapt maximum mean discrepancy on multiple layers to enforce the distribution similarity between source and target features. Deep CORAL (Sun & Saenko, 2016) uses feature covariance to measure the domain discrepancy. Another stream of works tries to enlarge the available train data space with augmentation of source domains (Carlucci et al., 2019; Dou et al., 2019; Qiao et al., 2020; Shankar et al., 2018; Zhou et al., 2020a;b). Several approaches leverage regularization through domain adversarial learning (Jia et al., 2020; Rahman et al., 2020) to address DG. Despite the promising results achieved by current domain generalization (DG) methods, all of them assume the availability of labeled or unlabeled target samples to aid in fine-tuning the models. However, our motivation and setting differ significantly. We strive to enhance the generalization ability of FER methods while maintaining their high classification accuracy. Besides, we exclusively train FER models on a single FER dataset and evaluate it on various unseen FER test sets. We refrain from accessing any target domain samples, rendering existing domain adaptation methods infeasible for our task.

## 3 PROBLEM DEFINITION

In this paper, we focus on improving the generalization ability of FER methods while maintaining their high classification accuracy. Learning is conducted on one given FER train set, and then test samples from different FER test sets with domain gaps of the train set should be recognized on the fly, which is similar to the real-world deployment of FER models.

FER models are trained with $\mathcal{D}_{train} = \{(\mathbf{x}_i, y_i)\}_{i=1}^N$, where $\mathbf{x}_i$ is the $i$-th training image and $y_i \in Y = \{1, \ldots, L\}$ is the corresponding label, $N$ is the number of training samples and $L$ is the number of expression classes. Traditional FER models are evaluated on the test set $\mathcal{D}_{test} = \{(\mathbf{x}_i, y_i)\}_{i=1}^M$ that has no domain gap between the train set, $M$ is the number of test samples. However, the real-world test set $\mathcal{D}_{real} = \{(\widetilde{\mathbf{x}}_i, y_i)\}_{i=1}^M$ is different from $\mathcal{D}_{test}$, as $\mathcal{D}_{real}$ might contain samples with domain gaps between the training samples. In this paper, we aim to train the FER model on $\mathcal{D}_{train}$ that can generalize well on the real-world test set $\mathcal{D}_{real}$. The difference between our task and the traditional FER is that we test the FER models on $\mathcal{D}_{real}$ instead of $\mathcal{D}_{test}$. The difference between our task and domain adaptation FER is that we do not have access to the unlabeled target samples from $\mathcal{D}_{real}$ to help fine-tune the FER model, which is more similar to the real-world setting as we cannot previously know the distribution of the target samples.

## 4 METHOD

CLIP generalizes well across a wide range of downstream tasks while it achieves relatively low classification accuracy on FER test sets compared with SOTA FER methods as the extracted features of CLIP contain many expression-unrelated features like age, gender, race, etc. SOTA FER methods achieve high classification accuracy while they fail to generalize to unseen test sets. We design a novel method to adapt CLIP for FER and combine the advantages of both CLIP and SOTA FER methods to achieve high classification accuracy and high generalization ability at the same time.

Specifically, we first use fixed CLIP to extract face features. Since CLIP is trained using large-scale image-text pairs, we can assume that the extracted face features generalize well across different domains of FER datasets. However, CLIP's extracted features contain many expression-unrelated features. Therefore, we have designed a novel method to adapt CLIP for FER. We train a FER model to generate sigmoid masks, which select expression-related features from the fixed face features extracted from CLIP. To further enhance the generalization ability of the learned masks, we introduce a channel-separation module and a channel-diverse loss to maximize mask diversity. It's worth noting that our pipeline is similar to how humans perceive expressions. As humans, when we encounter test samples with domain gaps from the training samples, we first extract their face features to exclude

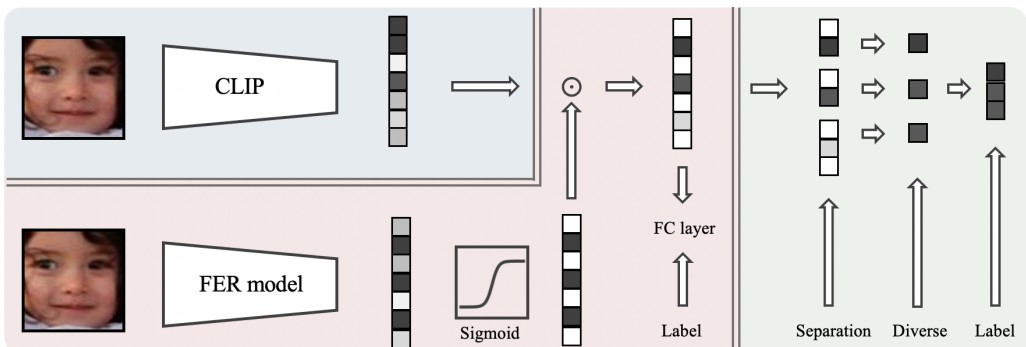

Figure 2: The framework of our proposed method. We adapt CLIP for FER to balance the classification accuracy and the generalization ability. CLIP is fixed to extract only image features of the training images, while the FER model is trained to learn masks for the fixed face features to extract only the expression-related features. It's worth noting that this process is similar to how humans perceive expressions: we first observe faces and then extract expression-relevant features. The learned masks are regularized by a sigmoid function to prevent overfitting. We further introduce a channel-separation module and a channel-diverse loss to make the learned masks as diverse as possible.

the domain features like image styles and backgrounds. Afterward, we focus solely on the features related to the expressions in order to determine the expression contained within the test sample. Following a similar approach, our proposed method initially utilizes CLIP to extract face features of test samples from all different domains. Then, a trained FER model generates sigmoid masks for selecting expression-related features from the face features and making decisions solely based on the selected features.

### 4.1 SIGMOID MASKS ON FIXED FACE FEATURES

Given images $\mathbf{x}$ from $\mathcal{D}_{train}$, we first extract the image features using CLIP, denoted as $\mathbf{F} \in \mathbb{R}^{N \times C}$, where $N$ is the number of images and $C$ the number of feature dimensions, we fix $\mathbf{F}$ during the training process to prevent the FER model to optimize face features to overfit the given train set. This operation improves the generalization ability of our proposed method. The FER model, such as ResNet-18, is trained to learn masks for the given face features. We first extract features $\mathbf{f} \in \mathbb{R}^{N \times C \times 1 \times 1}$ from $\mathbf{x}$ after the global average pooling (GAP) layer and resize them to generate the masks $\mathbf{M} \in \mathbb{R}^{N \times C}$ for face features. Further, in order to regularize the learned masks to generalize to unseen test samples, we apply a sigmoid function on $\mathbf{M}$ and get $\mathbf{M}_s$ as

$$\mathbf{M}_s = Sigmoid(\mathbf{M}). \tag{1}$$

The sigmoid function is vital to the success of our method as it introduces non-linearity into the model, which is crucial for capturing and learning non-linear patterns to generate the masks. The sigmoid function also normalizes $\mathbf{M}$, ensuring that the values of $\mathbf{M}$ fall within [0, 1], which reduces the overfitting ability of the learned masks as it reduces the range of variation of $\mathbf{M}$. Furthermore, the sigmoid function provides a probability-like output, where the output value represents the probability of selecting the feature of the corresponding channel, which is semantically similar to humans choosing expression-related features from the face features. The performance of our method without the sigmoid function drops drastically, which is shown in the appendix A.5.

We further utilize $\mathbf{M}_s$ to select the face features for expression recognition as

$$\widetilde{\mathbf{F}} = \mathbf{M}_s \mathbf{F}. \tag{2}$$

The classification loss is computed between the selected features $\widetilde{\mathbf{F}}$ and labels $y$ following

$$l_{cls} = -\frac{1}{N} \sum_{i=1}^{N} (\log \frac{e^{\mathbf{W}_{\mathbf{y}_i} \widetilde{\mathbf{F}}_i}}{\sum_{j}^{L} e^{\mathbf{W}_j \widetilde{\mathbf{F}}_i}}), \tag{3}$$

where $\mathbf{W}_{\mathbf{y}_i}$ is the $\mathbf{y}_i$-th weight from the FC layer and $\mathbf{y}_i$ is the label of the $i$-th image.

## 4.2 Channel-separation module and channel-diverse loss

To make the learned mask generalizable to other unseen FER test sets, we further design a channel-separation module to regularize the learned mask. Specifically, we set apart the masked features into seven pieces according to the channel dimension to make the masked features correspond to seven basic expressions and then max pool each piece to directly map them into logits. In such a design, we could avoid the use of the FC layer and directly connect the masked features to the FER labels. The motivation of our design is twofold: Firstly, the learning ability of the FC layer might be too strong to overfit the train set. Thus, we abandon the FC layer and directly transform the masked features into logits to prevent the FC layer from overfitting the labels with the learned features. Secondly, the channel size of 512 when using ResNet-18 might be too large to learn generalizable masks and could also lead to overfitting on the training set. If we set apart the mask into seven pieces and make each piece of the mask correspond to one basic expression, the mask piece with a small channel size will be more likely to only focus on the useful information.

Specifically, given the masked features $\widetilde{\mathbf{F}}$, we divide them according to the channel number $C$ to $L$ pieces corresponding to the class number $L$, $\widetilde{\mathbf{F}} = \{\widetilde{\mathbf{F}}_1, \widetilde{\mathbf{F}}_2, ..., \widetilde{\mathbf{F}}_L\}$. If $C$ can not be divided by $L$, we could divide the selected features non-uniformly. For example, when using ResNet-18, the channel size is 512, we could assign 73 channels for each of the 6 expressions and leave the rest 74 channels for the 'neural' expression. We utilize channel-dropping on the selected features to mitigate the overfitting problem of the selected features. The drop mask is denoted as $\mathbf{M}_{drop} = \{\mathbf{M}_1, \mathbf{M}_2, ..., \mathbf{M}_L\}$ corresponding to the selected features. Each mask contains 0 or 1 for feature selection. For example, in the $\mathbf{M}_1$, which is a vector of size $(73, 1)$, there are 10 channels randomly selected as 0 and all the rest are 1, leading to a drop rate of 10/73. The channel drop module guides the model to focus on all channels, which increases the generalization ability of the masked features. The channel-drop rate only slightly influences the performance at a reasonable rate, and we maintain the drop rate at 10/73 across all our experiments, as we consider it a minor trick rather than one of our contributions. After channel dropping, the selected features are denoted as

$$\mathbf{M}_{sel} = \widetilde{\mathbf{F}}\mathbf{M}_{drop} = \{\widetilde{\mathbf{F}}_1\mathbf{M}_1, \widetilde{\mathbf{F}}_2\mathbf{M}_2, ..., \widetilde{\mathbf{F}}_L\mathbf{M}_L\}. \tag{4}$$

The selected features $\mathbf{M}_{sel}$ are downsized to logits of size $(N, L)$ for classification through a max-pooling operation.

$$\widetilde{\mathbf{M}} = \{max(\widetilde{\mathbf{F}}_1\mathbf{M}_1), max(\widetilde{\mathbf{F}}_2\mathbf{M}_2), ..., max(\widetilde{\mathbf{F}}_L\mathbf{M}_L)\}. \tag{5}$$

Then, we compute a classification loss $l_{sep}$ with the labels and the logits $\widetilde{\mathbf{M}}$ obtained by the separation module without the FC layer.

$$l_{sep} = -\frac{1}{N}\sum_{i=1}^{N}(\log \frac{e^{\mathbf{W}_{y_i}\widetilde{\mathbf{M}}_i}}{\sum_{j}^{L} e^{\mathbf{W}_j\widetilde{\mathbf{M}}_i}}), \tag{6}$$

To increase the generalization ability of the learned masks, we want to make the channels corresponding to each class as diverse as possible. Thus, we further introduce a channel-diverse loss. Specifically, we input the $\widetilde{\mathbf{F}}$ into the max pooling operation to get

$$\widetilde{\mathbf{F}}_{max} = \{max(\widetilde{\mathbf{F}}_1), max(\widetilde{\mathbf{F}}_2), ..., max(\widetilde{\mathbf{F}}_L)\}, \tag{7}$$

the selected max feature channel is regularized to be diverse with other feature channels by the channel-diverse loss $l_{div}$,

$$l_{div} = 1 - \frac{1}{Nc}\sum_{i=1}^{N}\sum_{j=1}^{L}\widetilde{\mathbf{F}}_{max}, \tag{8}$$

where $c$ is the number utilized for normalization, we experimentally set $c$ as 73, which is the same as the channel number of the separated piece. The channel-diverse loss regularizes the max value of each piece of $\widetilde{\mathbf{F}}_{max}$ as large as possible, which separates the largest value of each piece of $\widetilde{\mathbf{F}}_{max}$ from other values, making the values of the channel dimension more diverse.

Combining the separation loss and channel-diverse loss, we aim to learn a powerful while generalizable mask to only select useful expression features from the fixed generalizable face features. The total train loss is summarized as

$$l_{train} = l_{cls} + \lambda l_{sep} + \beta l_{div}. \tag{9}$$

Table 1: The test accuracy of different FER methods on various FER test sets. The FER model is exclusively trained using the dataset in the second column, and we evaluate it on all five test sets. Our method significantly outperforms SOTA FER methods by large margins on nearly all unseen test sets and performs on par with them on the corresponding test set. We underline the best accuracy of other FER methods and highlight the improvement achieved by our method compared to it in blue.

| Method | RAF-DB | FERPlus | AffectNet | SFEW2.0 | MMA | Mean |
|---|---|---|---|---|---|---|
| Baseline | 88.40 | 58.05 | 43.25 | 42.76 | 42.61 | 55.01 |
| SCN | 87.32 | 58.37 | 42.85 | 44.89 | 36.52 | 53.99 |
| RUL | 88.66 | 57.89 | 43.82 | 46.91 | 37.11 | 54.88 |
| EAC | **89.15** | 56.33 | 44.02 | 42.76 | 37.95 | 54.04 |
| Ours | 88.72 | **73.16** (+14.79) | **45.86** (+1.84) | **52.86** (+5.95) | **56.80** (+14.19) | **63.48** |

| Method | FERPlus | RAF-DB | AffectNet | SFEW2.0 | MMA | Mean |
|---|---|---|---|---|---|---|
| Baseline | 88.17 | 56.23 | 36.31 | 45.45 | 59.85 | 57.20 |
| SCN | 86.80 | 68.71 | 32.42 | 43.10 | 59.12 | 58.03 |
| RUL | 88.40 | 51.89 | 35.88 | 45.90 | 58.00 | 56.01 |
| EAC | 89.03 | 41.62 | 36.49 | 45.79 | 59.89 | 54.56 |
| Ours | **89.51** | **72.91** (+4.20) | **39.44** (+2.95) | **49.38** (+3.48) | **60.14** (+0.25) | **62.28** |

| Method | AffectNet | RAF-DB | FERPlus | SFEW2.0 | MMA | Mean |
|---|---|---|---|---|---|---|
| Baseline | 56.06 | 71.22 | 66.75 | 44.44 | 44.35 | 56.56 |
| SCN | 62.48 | 70.70 | 63.98 | 41.98 | 38.51 | 55.53 |
| RUL | 58.70 | 55.83 | 52.88 | 34.01 | 31.93 | 46.67 |
| EAC | **63.77** | 66.10 | 57.19 | 44.89 | 33.49 | 53.09 |
| Ours | 57.87 | **72.69** (+1.47) | **69.94** (+3.19) | **51.18** (+6.29) | **49.65** (+5.30) | **60.27** |

| Method | SFEW2.0 | RAF-DB | FERPlus | AffectNet | MMA | Mean |
|---|---|---|---|---|---|---|
| Baseline | 39.68 | 46.68 | 33.41 | 29.18 | 22.85 | 34.36 |
| SCN | 43.39 | 46.74 | 32.58 | 23.19 | 17.38 | 32.66 |
| RUL | 42.00 | 46.35 | 36.02 | 30.41 | 22.06 | 35.37 |
| EAC | 43.39 | 47.29 | 33.66 | 25.57 | 22.26 | 34.43 |
| Ours | **45.01** | **54.43** (+7.14) | **48.39** (+12.37) | **32.24** (+1.83) | **36.34** (+13.49) | **43.28** |

| Method | MMA | RAF-DB | FERPlus | SFEW2.0 | AffectNet | Mean |
|---|---|---|---|---|---|---|
| Baseline | 61.93 | 70.50 | 69.97 | 45.79 | 37.46 | 57.13 |
| SCN | 63.00 | 74.09 | **73.99** | 45.90 | 35.94 | 58.58 |
| RUL | 61.70 | 71.94 | 69.05 | 39.21 | 34.45 | 55.27 |
| EAC | 65.06 | 74.32 | 71.85 | 42.87 | 35.83 | 57.99 |
| Ours | **65.97** | **78.36** (+4.04) | 73.57 (-0.42) | **49.05** (+3.15) | **41.85** (+4.39) | **61.76** |

After training, the module to compute $l_{sep}$ and $l_{div}$ can be abandoned, we only need to keep the module that is used to compute $l_{cls}$ to recognize the test samples as $l_{sep}$ and $l_{div}$ are only used to regularize the mask learning during training.

## 5 EXPERIMENTS

### 5.1 DATASETS AND METHODS

RAF-DB (Li et al., 2017) is annotated with seven basic expressions by 40 trained human coders, including 12,271 images for training and 3,068 images for testing.

FERPlus (Barsoum et al., 2016) is extended from FER2013 (Goodfellow et al., 2013) with cleaner labels, which consists of 28,709 training images and 3,589 test images collected by the Google search engine, we utilize the same seven basic expressions with the RAF-DB.

AffectNet (Mollahosseini et al., 2017) is a large-scale FER dataset, which contains eight expressions (seven basic expressions and contempt). There are a total of 286,564 training images and 4,000 test images. We utilize the seven basic expressions in our experiments.

SFEW2.0 is the most commonly used version of SFEW (Dhall et al., 2011). SFEW2.0 contains 958 train samples, 436 validation samples, and 372 test samples. Each image is assigned to one of the seven basic expressions.

MMA is a large-scale FER dataset with the majority of expressions from individuals of European and American descent. The dataset contains 92,968 training samples, 17,356 validation samples, and 17,356 test samples. Each image is assigned to one of the seven basic expressions.

Since we aim to evaluate the classification accuracy and generalization ability of FER methods, our main comparison methods are SOTA FER methods like SCN (Wang et al., 2020), RUL (Zhang et al., 2021) and EAC (Zhang et al., 2022). We only use one train set to train the FER model, in the test phase, different unseen FER test sets are utilized to simulate the real-world deployment. We do not compare with domain adaptation FER methods as we do not use any unlabeled samples of the target domain, which makes existing domain adaptation methods in FER impractical.

## 5.2 IMPLEMENTATION DETAILS

We utilize ResNet-18 (He et al., 2016) as the backbone in most experiments of our paper. The learning rate $\eta$ is set to $0.0002$ and we use Adam (Kingma & Ba, 2014) optimizer with weight decay of $0.0001$. We utilize a learning rate scheduler of ExponentialLR, with a gamma of $0.9$. We set the weight for the channel-wise loss as $1.5$ and the weight for the diverse loss as $5$. The max training epoch $T_{max}$ is set to $60$.

## 5.3 MAIN EXPERIMENTS

To evaluate the classification accuracy and generalization ability of existing FER methods, we utilize one of the five FER datasets as the training set. Instead of testing only on the test set of the corresponding train set, we test on all five FER test sets. To simulate the real-world deployment, we do not have access to any images of the different FER test sets during the training stage, which means the domain adaptation method is not feasible under our setting. The results shown in Table 5 demonstrate that SOTA FER methods do not perform well on FER test sets with domain gaps between the train set. For example, though the SOTA FER method EAC improves the baseline method by remarkable margins on the test set corresponding to the train set, it achieves similar or even inferior results on other FER test sets than the baseline method. We speculate the reason might lie in that EAC fits the train set too much to learn generalizable FER features to other different FER test sets. Our method maintains high accuracy on the test set corresponding to the train set. Under other unseen test sets, we underline the best result of other FER methods and compare it with our method. Our method outperforms SOTA FER methods by large margins almost under all settings. We also show the mean accuracy on five FER test sets on the right. Under different FER train sets, our method always achieves the best mean accuracy on five FER test sets. We further provide the test accuracy of each expression class and the mean accuracy on all expression classes in Table 6 and find that our method also achieves the best mean accuracy and the best accuracy on most of the expression classes.

## 5.4 ABLATION STUDY

To study the effectiveness of each of the proposed modules in our method, we carry out thorough ablation studies. The results in Table 3 show that without our method, the FER model cannot generalize to other datasets that have domain gaps with the train set, which is unsatisfactory for the real-world deployment of FER models. With our proposed sigmoid mask learning, the performance on other unseen test sets outperforms the baseline by a large margin. However, the fitting probability of the mask is too strong as the dimension of 512 might be too much to learn generalizable expression features. Thus, we introduce the channel-separation module which separates the masked features into pieces corresponding to the different expression classes. The performance improves from only using the sigmoid mask module. We introduce a channel-diverse loss to make the channels in each piece as diverse as possible, which further improves the accuracy. From the results in Table 3, each

Table 2: The performance on the FERPlus test set with the accuracy of each expression class, when the train set is RAF-DB. Overall accuracy is the accuracy on the whole test set, mean accuracy is the mean value of the accuracy of each expression class. Our method achieves both the best overall and the mean accuracy compared with other methods.

| Method | Surprise | Fear | Disgust | Happiness | Sadness | Anger | Neutral | Overall | Mean |
|---|---|---|---|---|---|---|---|---|---|
| Baseline | 61.36 | 24.10 | 22.22 | 83.43 | 68.49 | 49.45 | 37.71 | 58.05 | 49.54 |
| SCN | **82.83** | 14.46 | 27.78 | 77.72 | 61.20 | **61.17** | 35.78 | 58.37 | 51.56 |
| RUL | 69.70 | **45.78** | 22.22 | 82.53 | 70.57 | 52.01 | 31.93 | 57.89 | 53.54 |
| EAC | 70.71 | 39.76 | 16.67 | 81.30 | 76.04 | 58.24 | 25.14 | 56.33 | 52.55 |
| Ours | 78.54 | 37.35 | **33.33** | **94.96** | **78.65** | 57.51 | **58.72** | **73.16** | **62.72** |

Table 3: The ablation study of our proposed method. The results on different FER test sets show that the most effective module of our method is the sigmoid mask module. The channel-separation module and the channel-diverse loss make the learned mask more generalizable to other FER test sets and further improve the performance of our method.

| Mask | Separation | Diverse | FERPlus | AffectNet | SFEW2.0 | MMA | Mean |
|---|---|---|---|---|---|---|---|
| | | | 58.05 | 43.25 | 42.76 | 42.61 | 46.67 |
| ✓ | | | 70.90 | 43.77 | 51.63 | 55.65 | 55.49 |
| ✓ | ✓ | | 72.01 | 45.17 | **53.31** | 56.69 | 56.80 |
| ✓ | ✓ | ✓ | **73.16** | **45.86** | 52.86 | **56.80** | **57.17** |

module contributes to the success of our proposed method, and combining them together achieves the best result.

## 5.5 DIFFERENT BACKBONES

We combine our method with different backbones to show its generalization ability. The results in Table 4 illustrate that our method improves the performance of baseline under different backbones by remarkable margins. Specifically, the improvement is largest when using ResNet-18 as the backbone, the reason might lie in that the dimension of the output feature of ResNet-18 is 512, which is the same as the dimension of the output feature of CLIP. When using MobileNet (Howard et al., 2017) or ResNet-50, we reduce the size of the output feature through mean operation to suit the feature dimension of CLIP, which might slightly limit the performance improvement. For example, when the backbone is ResNet-50, we simply reduce the output feature dimension of 2048 to 512 through mean operation using sliding windows. We also observe that stronger backbones have better generalization ability in our experiment. The ResNet-50 backbone achieves the overall best performance across experiments.

## 5.6 HYPERPARAMETER STUDY

We study the influence of the weight of separation loss $\lambda$ and the weight of diverse loss $\beta$ on our method. The results shown in Figure 3 illustrate that both the $\lambda$ and $\beta$ can be chosen from a wide range and the performance is only slightly different across an order of magnitude, e.g., $\lambda$ from $[0.5, 5]$ and $\beta$ from $[1, 10]$. For simplicity, we choose $\lambda$ as 1.5 and $\beta$ as 5 across all our experiments.

## 5.7 VISUALIZATION

We visualize the learned features using t-SNE (van der Maaten & Hinton, 2008) to provide an intuitive understanding of our method in Fig. 4. The FER model is trained on RAF-DB train set and the features on the FERPlus test set are displayed.

Table 4: The performance of our method under different backbones.

| Backbone | RAF-DB | FERPlus | AffectNet | SFEW2.0 | MMA | Mean |
|---|---|---|---|---|---|---|
| MobileNet | 84.65 | 61.33 | **43.45** | 43.21 | 40.67 | 54.66 |
| MobileNet + Ours | **85.07** | **64.97** | 42.91 | **45.45** | **43.11** | **56.30** |
| ResNet-18 | 88.40 | 58.05 | 43.25 | 42.76 | 42.61 | 55.01 |
| ResNet-18 + Ours | **88.72** | **73.16** | **45.86** | **52.86** | **56.80** | **63.48** |
| ResNet-50 | 88.49 | 70.13 | 47.46 | 49.94 | 48.90 | 60.98 |
| ResNet-50 + Ours | **89.05** | **75.26** | **47.49** | **51.07** | **57.46** | **64.07** |

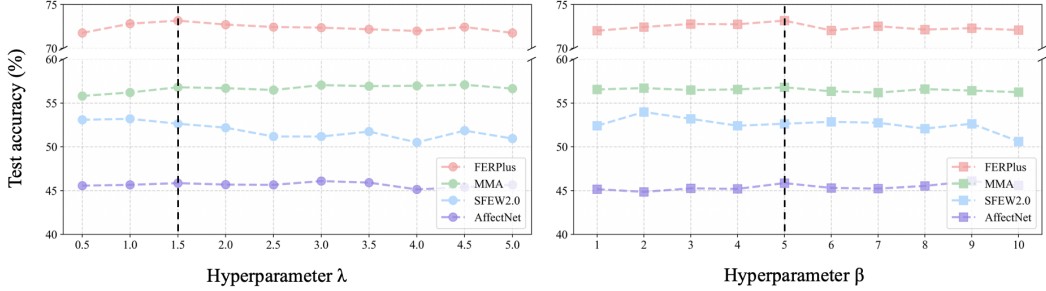

Figure 3: The hyperparameter study of our method. Our method is not very sensitive to the two hyperparameters, and we could choose them from a wide range. For simplicity, we use $\lambda = 1.5$ and $\beta = 5$ across all experiments.

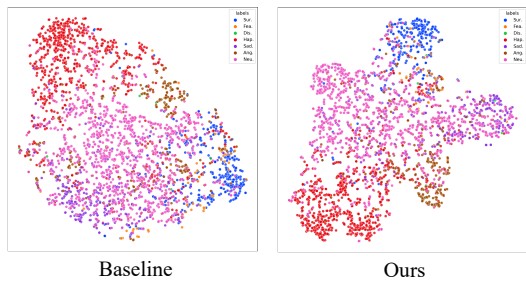

Baseline            Ours

Figure 4: The t-SNE visualization of the learned features of the baseline and our method.

The learned features of the baseline are extracted after the global average pooling layer. The learned features of the FER model under our method are the learned masks, which are the outputs of the FER model, corresponding to the features extracted from the baseline. We can observe that in Fig. 4, the features on FERPlus are not well separated by the baseline method. The reason lies in that there is a domain gap between RAF-DB and FERPlus, which means that the baseline method cannot generalize well to other test sets different from the train set of RAF-DB. Our method outperforms the baseline method and separates the samples from different expression classes better.

## 6 CONCLUSION

In this paper, we study the balance between classification accuracy and the generalization ability of FER methods. We observe that existing FER methods fail to generalize on test sets with domain gaps between the training set, which is far from reliable in real-world deployment. To address this problem, we design a novel method to learn sigmoid masks on fixed CLIP face features to adapt CLIP for FER. To further enhance the generalization ability on unseen FER datasets, we propose a channel-separation module and a channel-diverse loss to make the learned sigmoid masks as diverse as possible. Extensive experiments on different FER datasets and backbones show that our method outperforms state-of-the-art FER methods by remarkable margins and achieves both high classification accuracy and high generalization ability.

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

# A  APPENDIX

## A.1  PIPELINE OF OUR METHOD

We summarize the pipeline of our method in Alg. 1. We utilize fixed pre-trained large model like CLIP $f_{\text{fixed}}$ to extract generalizable face features for any given facial expression recognition (FER) samples.We then train a newly initialized FER model $f$ from scratch to extract useful expression-related features from the given face features. We assume that if the FER model learns how to extract expression features from given face features, it can generalize to other unseen FER domains based on the generalizable face features extracted from CLIP. We input the learned mask of the FER model to a sigmoid function to reduce overfitting. The learned sigmoid mask is multiplied with the fixed face features to select useful expression-related features. To make the selected features generalizable, we separate them according to the channel dimension into seven pieces corresponding to the seven basic expression classes. Each piece of the selected features represents the features of a certain expression class. To further increase the generalization ability, we make the channels within each piece as diverse as possible through a channel-diverse loss. The training loss is calculated by summing the classification loss, channel-separation loss, and the channel-diverse loss.

---

**Algorithm 1** Training Algorithm

---

**Require:** Fixed pre-trained large model $f_{\text{fixed}}$, FER dataset $D$, newly initialized FER model $f$ with parameters $\Theta$, learning rate $\eta$, total epochs $T_{\max}$, data loader iterations $I_{\text{train}}$.
1: **for** $t = 1$ to $T_{\max}$ **do**
2:    **for** $n = 1$ to $I_{\text{train}}$ **do**
3:       **Fetch** mini-batch $D_n$ from $D$
4:       **Extract** fixed face features $\mathbf{F}$ using $f_{\text{fixed}}$
5:       **Extract** facial expression features $\mathbf{f}$ using $f$
6:       **Calculate** sigmoid mask $\mathbf{M}$ by resizing $\mathbf{f}$ and inputting it to the sigmoid function
7:       **Obtain** the masked feature $\tilde{\mathbf{F}}$ using Eq. (2)
8:       **Calculate** the classification loss $\ell_{cls}$ using Eq. (3)
9:       **Separate** the masked feature $\tilde{\mathbf{F}}$ and apply channel dropping as per Eq. (4)
10:      **Calculate** the logits without FC layer using Eq. (5)
11:      **Calculate** the channel-separation loss $\ell_{sep}$ using Eq. (6)
12:      **Separate** and max pool the masked feature $\tilde{\mathbf{F}}$ according to Eq. (7)
13:      **Calculate** the channel-diverse loss $\ell_{div}$ using Eq. (8)
14:      **Calculate** the training loss $\ell_{train}$ using Eq. (9)
15:      **Update** $\Theta = \Theta - \eta \overline{\nabla} \ell_{train}$
16:    **end for**
17: **end for**
**Ensure:** The trained FER model $f$, which can selectively extract expression-related features from the given fixed face features.

---

## A.2  IMPLEMENTATION DETAILS ON AFFECTNET

Implementation details are slightly different on AffectNet dataset as it is large-scale and imbalanced. The learning rate is 0.0001 and the gamma of the scheduler is 0.8. As AffectNet is extremely imbalanced, we adopt a balanced sampler to keep the samples of each class similar within each batch. The training epoch is 20 instead of 60 because AffectNet has much more training samples than RAF-DB.

## A.3  PERFORMANCE WITHOUT PRE-TRAINED FER MODEL

To further illustrate the effectiveness of our proposed method, we carry out experiments without using the pre-trained backbone of FER model.

All the FER methods are trained from scratch without a pre-trained backbone on RAF-DB and tested on all five different FER test sets. The results are shown in Table 5. We observe that our method outperforms the SOTA FER methods on the generalization ability by even larger margins

| Method | RAF-DB | FERPlus | AffectNet | SFEW2.0 | MMA | Mean |
|---|---|---|---|---|---|---|
| With Pre-Trained FER Backbone | | | | | | |
| Baseline | 88.40 | 58.05 | 43.25 | 42.76 | 42.61 | 55.01 |
| SCN | 87.32 | 58.37 | 42.85 | 44.89 | 36.52 | 53.99 |
| RUL | 88.66 | 57.89 | 43.82 | 46.91 | 37.11 | 54.88 |
| EAC | **89.15** | 56.33 | 44.02 | 42.76 | 37.95 | 54.04 |
| Ours | 88.72 | **73.16** (+14.79) | **45.86** (+1.84) | **52.86** (+5.95) | **56.80** (+14.19) | **63.48** |
| Without Pre-Trained FER Backbone | | | | | | |
| Baseline | 75.01 | 36.75 | 24.48 | 19.75 | 19.38 | 35.07 |
| SCN | 75.23 | 38.48 | 21.41 | 17.17 | 30.11 | 36.48 |
| RUL | 79.53 | 35.86 | 16.42 | 11.78 | 11.89 | 31.10 |
| EAC | 80.64 | 41.73 | 23.73 | 22.22 | 29.51 | 39.57 |
| Ours | **85.92** | **69.14** (+27.41) | **40.96** (+16.48) | **45.79** (+23.57) | **54.65** (+24.54) | **59.29** |

Table 5: Comparison between with or without pre-trained FER backbone. The test accuracy of different FER methods on various FER test sets is shown. We underline the best accuracy of other FER methods and highlight the improvement achieved by our method compared to it in blue. Our method outperforms SOTA FER methods by even larger margins when without a pre-trained FER backbone.

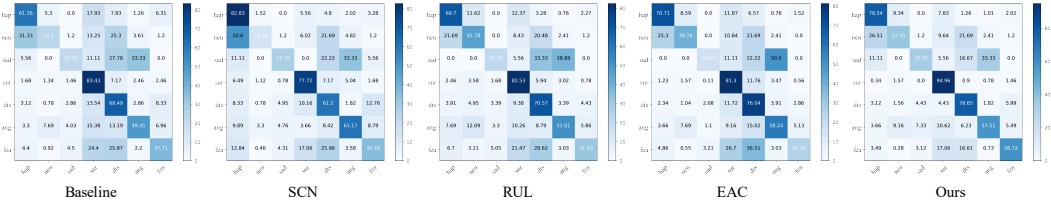

Figure 5: The confusion matrices of different methods when the train set is RAF-DB and the test set is FERPlus. Zoom in for better visualization.

when without a pre-trained FER backbone. For example, without a pre-trained FER backbone, our model increases the second-best performance on FERPlus by 27.41% compared with 14.79% when with the pre-trained FER backbone. The performance of all FER methods drops when without the pre-trained FER backbone, however, our method achieves a very similar performance between the two groups, only decreasing the mean performance from 63.48% to 59.29%, which is acceptable. While the EAC method decreases the mean performance from 54.04% to 39.57%, which is rather drastic. The results illustrate that the pre-trained FER backbone is not necessary for our method to achieve the best performance. Our method without a pre-trained FER backbone still achieves a mean accuracy of 59.29%, outperforming the best mean accuracy of other methods with a pre-trained FER backbone of 55.01%. From the comparison between the two groups, we conclude that our method is robust and performs well even without the pre-trained FER backbone. The reason lies in that our method only learns a mask to select fixed face features instead of learning the whole expression features, which does not need a very strong FER model to perform well. Furthermore, a simple FER model without pre-training can extract simple patterns to select fixed face features, which is more likely to generalize across different FER datasets.

## A.4 CONFUSION MATRICES

Due to the space limitation, we display the accuracy of each class on FERPlus in Table 2 of our paper. In this section, to provide a more accurate reference, we display the confusion matrices of different methods in Fig. 5. From the results, we conclude that our method achieves the best accuracy in most of the expression classes.

| Method | RAF-DB | FERPlus | AffectNet | SFEW2.0 | MMA | Mean |
|---|---|---|---|---|---|---|
| No sigmoid | **89.24** | 62.58 | **46.37** | 46.58 | 45.49 | 58.05 |
| Ours | 88.72 | **73.16** (+10.58) | 45.86 (-0.51) | **52.86** (+6.28) | **56.80** (+11.31) | **63.48** |

Table 6: Influence of the sigmoid function on our method. Our method uses the sigmoid mask to select expression-related features, the comparison group has no sigmoid function and the others are the same as our method. The results show that the sigmoid function is very important for our method to learn generalizable masks.

## A.5 THE EFFECT OF THE SIGMOID FUNCTION

To illustrate the effectiveness of the sigmoid function applied to the learned mask, we design a comparison group of our method without the sigmoid function, while we keep the others exactly the same. The experiment results on different FER test sets are shown in Table 6. The results demonstrate that the sigmoid function for mask learning is very important, without it, our method barely works. The reason lies in that the sigmoid function normalizes the learned masks, ensuring that the values fall within [0, 1], which reduces the overfitting ability of the learned masks.

