# OpenReview forum: "CLIP Facial Expression Recognition: Balancing Precision and Generalization"
_ICLR.cc/2024/Conference — ICLR 2024 Conference Withdrawn Submission_

### Official Review · Reviewer_GoGn · 2023-10-30

**Soundness:** 3 good
**Presentation:** 4 excellent
**Contribution:** 4 excellent
**Rating:** 6
**Confidence:** 5

**Summary:**

This paper investigates the trade-off between classification accuracy and generalization ability in the facial expression recognition (FER) field, which has been less touched before. It finds that high classification accuracy comes along with overfitting of the given domain, which leads to failure of generalization on unseen different FER domains. It proposes to learn sigmoid masks to select fixed face features extracted by CLIP for expression recognition, which combines the generalization ability of CLIP and the high classification accuracy of traditional FER models. It further proposes a channel-separation module and a channel-diverse module to make the masks more diverse to improve the generalization ability. Experiments on five FER datasets validate the proposed method is superior to other SOTA FER methods.

**Strengths:**

+ This paper investigates a new problem, which is different from existing FER cross-domain adaptation methods. Existing cross-domain methods in FER all require unlabeled target samples to help tune the FER model trained on the source domain. However, the proposed method does not require unlabeled target domain samples to tune the FER model, which can directly generalize to other unseen FER domains.

+ The method is novel to me. I find it interesting that the authors learn sigmoid masks on the fixed face features extracted from CLIP to select expression features, which could combine the generalization ability of CLIP and the high classification accuracy of the FER model. The design of the sigmoid mask is novel and is vital to the success of the proposed method. I noticed that in the Appendix, without using the sigmoid function, the learned mask fails to improve the generalization ability of the FER model.

+ Experiments on five FER datasets show that the proposed method outperforms other SOTA methods by clear margins. As claimed by the authors, existing SOTA FER methods can work well on the given test set while achieve low classification accuracy on unseen test sets. The proposed method maintains the high classification accuracy on the given test set, while at the same time improves the generalization ability on other unseen test sets, which is certainly superior to the SOTA FER methods.

**Weaknesses:**

- Some other design options should be studied. For example, in the channel separation module, the authors set apart the channels evenly into 7 pieces. The authors should discuss more options as different expression classes have different difficulties. For example, it is well known that the happy expression has the most training sample and it is easy to be captured, while things are different when it comes to the fear or disgust expression as fear is easy to be confused with surprise and disgust is very subtle and they both have very few training samples. Thus, learning to capture fear and disgust is more difficult than learning to capture happy. I am curious what will happen if we assign difficult classes like fear or disgust more channels.

- I noticed the authors study the mask w/wo sigmoid function in the Appendix. However, I think this is a very important experiment as the sigmoid function is vital for the success of the proposed method and I think it is a novel design. Thus, I strongly suggest the authors include the corresponding experiments in the main paper instead of putting them in the Appendix.

- Some experiments are missing. I suggest the authors incorporate experiment results pertaining to the utilization of different channel drop rates.

**Questions:**

I have carefully examined Table 1 and observed a compelling trend wherein the proposed method demonstrates a notable superiority over other methods, particularly when applied to unseen FER datasets while leveraging RAF-DB as the training dataset. It is also intriguing to note that its performance margin relative to other methods is comparatively lower when tested on the AffectNet dataset. Could the authors explain that?

---

### Official Review · Reviewer_Re8m · 2023-10-30

**Soundness:** 3 good
**Presentation:** 3 good
**Contribution:** 2 fair
**Rating:** 3
**Confidence:** 5

**Summary:**

The authors proposed to utilize CLIP visual features to enhance the quality of facial expression recognition (FER) model in cross-dataset settings. They use three components for loss function: 1) cross-entropy loss for classification; 2) separation of features selected for every emotional class; and 3) diversity of channels using max pooling. Experiments with 5 datasets of static facial photos (AffectNet, RAF-DB, FERPlus, SFEW and MMA) demonstrate that the proposed approach achieves rather high accuracy in cross-dataset scenario.

**Strengths:**

1.	Interesting idea of channel-separation and channel diverse losses to combine CLIP and convolutional neural networks for FER
2.	Extensive experimental study of the proposed model for several datasets

**Weaknesses:**

1.	Though the results in cross-dataset settings (Table 1) are slightly better than for competing methods, they are still much worse (5-40%) when compared to training on the same dataset. It seems that the proposed method cannot solve the main issue of existing FER models.
2.	The authors do not compare their technique with the state-of-the-art models for each dataset. Just a brief look at, e.g., AffectNet (https://paperswithcode.com/sota/facial-expression-recognition-on-affectnet) clearly shows that the authors’ results are much worse: 7-class accuracy is 57.87%, which is much worse than the top result (67.49%). The same comparison should be made for other datasets. I expect the authors to support their claim “Current facial expression recognition (FER) methods excel in achieving high classification accuracy but often struggle to generalize effectively across various unseen test sets” by showing the results of the state-of-the-art models trained on one dataset and tested on other datasets. The top model might be better on the same dataset but should show much worse results on other datasets. In fact, I personally used several EfficientNet models from hsemotion dataset and noticed that they are quite good in real settings.
3.	CLIP model is mainly used due to its zero-shot capabilities caused by multi-modal training and possibility to use textual prompts. It is questionable if the visual part of the CLIP model only is the best choice for the authors’ model. I believe there exist other visual neural networks pre-trained on very large datasets that may be also used here, so the constraint to use CLIP is really required.
4.	I cannot find the name of the CLIP model, though there exist several different ViT-based models. I believe it is important to show the experiments with different CLIP models.
5.	Mathematical notation is weird. For example, index j is not used in Eq. 8 in F_{max}. \tilde{M}_ i is undefined in Eq.6, M_{sel} is not used in Eq. 5, etc.

**Questions:**

1.	How to decrease the gap for your model trained on different datasets? I’m afraid that it’s impossible to use the model if it shows 40-45% accuracy on AffectNet with 7 classes…
2.	What are the results of existing state-of-the-art models trained on each dataset and tested on other datasets?
3.	Am I correct that textual prompts and language part of CLIP are not used as it is done in EmoCLIP (https://arxiv.org/pdf/2310.16640.pdf)? If so, is it possible to use other visual models instead of CLIP?
4. What is the overhead in running time when using the proposed model compared to a single neural network?

---

### Official Review · Reviewer_cU44 · 2023-10-30

**Soundness:** 2 fair
**Presentation:** 3 good
**Contribution:** 2 fair
**Rating:** 3
**Confidence:** 5

**Summary:**

This paper proposes a generaliable face expression recognition method based on the feature from CLIP and learned feature masks and selection strategy. Experimental results on multiple FER datasets show the effectiveness of the propsoed method.

**Strengths:**

This paper is well written and easy to follow, and the core idea of this paper is intutive.

**Weaknesses:**

Here are some questions about this paper:
1) The contribution of this paper is limited. The core idea of this paper is leveraging the rich representation of CLIP feature for expression recognition. However, since the CLIP feature is trained on large-scale open world data, the comparsion with existing SOTA methods are not fair.
2) It seems that the propsoed method can be easily adapted to other tasks such as face attributes or face recognition. More experiments are need to demonastrate the effectiveness of the method.
3) Please describe the baseline in all tables.
4) From Table 2, we can see that the major performance gain is from happiness and neutral class. This is reasonable because of the nature of CLIP features. However, the generability of a method should fouce more on the rare (tail) classes, for which the proposed method is not good at. Besides, the feature distribution of Fig. 4 is not seperatable for the baseline and proposed method.

**Questions:**

See Weaknesses above

---

### Official Review · Reviewer_zwTJ · 2023-11-02

**Soundness:** 3 good
**Presentation:** 3 good
**Contribution:** 3 good
**Rating:** 5
**Confidence:** 5

**Summary:**

This paper adapts the features from CLIP to the facial expression recognition(FER) task. The main idea is to learn a mask for the visual features from CLIP by the supervision of FER. Since the masks filter unrelated features, the masked features focus on the information about facial expressions and preserve the generalization ability of the original CLIP feature. Experiments demonstrate the advantages of the proposed method over other SOTA FER methods under cross-dataset protocol.

The proposed method is simple and effective. It provides a strategy to improve the feature's performance on downstream tasks and preserve the generalization ability at the same time. The paper could be improved if the intuition and insight of the method could be further explored and discussed (mentioned in weakness 4).

**Strengths:**

1. The proposed strategy that learn a mask on the general features. It provides a simple and effective approach to improve the performance on downstream task and preserve the generalization ability.

2. This paper well written and easy to follow.

3. Extensive experiments are conducted on AU detection task and facial expression recognition task. The experimental results show the advantage of the proposed method.

**Weaknesses:**

1. The illustrated channel-separation is confused in Fig.2. It is easy to mis-understand that the channels of the CLIP features are separated and each channel of the features is assigned to one emotion. However, it is difficult to determine which emotion category each channel should be assigned. The main text provides more clear details of how to mask the features and separate the channels. Therefore, please revise Fig. 2 to make it more clear.

2. Eq.7 and Eq.8 are not clear.
   - In Eq.7, \tilda F_max is an output of the max pooling of L max{F_i}. It indicates that each sample has one \tilda F_max instead of L \tilda F_max. However, Eq.8 accumulates N \times L \tilda F_max. Should Eq.8 accumulate by i from 1 to N, without by j from 1 to L?

3. What is the setting for the baseline method in the experiment? Is the baseline the one with the CLIP feature and FC classifier?

4. In the proposed method, the masks are different for different input samples. Does it indicate that, for different samples, the information of the emotion is conveyed in a different position to a different extent? However, the general feature is more likely to encode the same information in the same way, even if the input samples are different. Could the author explain why the same information would be encoded in different ways for different inputs? In my understanding, it is more reasonable to train a set of masks for each class in downstream tasks. The masks should be different for different classes, but be the same for different input faces. Could the author also provide discussion and experimental results in this setting?

**Questions:**

Please refer to the weakness.